# Awareness of Behavioural Intervention Strategies in Curbing Mental Health Issues among Youth in Malaysia

**DOI:** 10.3390/ijerph192215376

**Published:** 2022-11-21

**Authors:** Sharmini Gopinathan, Anisha Haveena Kaur, Lai Ming Ming, Mazni Binti Alias, Segaran Veeraya

**Affiliations:** 1Faculty of Management, Multimedia University, Cyberjaya 63000, Malaysia; 2Graduate School of Business, Asia Pacific University, Kuala Lumpur 57000, Malaysia

**Keywords:** youth mental health, mental health, early intervention, prevention, mental health services, public mental health, emerging adulthood, adolescence, innovative therapies, healthcare systems

## Abstract

Mental health is a growing concern among people worldwide. Mental health issues are one of the main factors contributing to adolescent health-related burden. Malaysia, in particular, has seen an increase in the number of youths facing mental health issues. The government aims to take action by promoting mental health well-being as well as providing care and recovery to those who are affected. This study aimed to examine measures that could potentially improve and curb mental health issues among youth in Malaysia by adopting the use of behavioural intervention technologies. Three underlying models of intervention were studied, namely, the internet intervention model, Fogg behaviour model, and persuasive system design. A total of 103 respondents between the ages of 18 to 23 years participated in the research survey, and the results revealed that mood changes and thoughts, feelings, and actions were the intervention strategies that showed a positive significance in the improvement of mental health among youth in Malaysia. Social distractions, peer motivation, ease of access to help, and sense of belonging and mindfulness did not show a positive significance when it came to behavioural intervention technologies used to improve mental health.

## 1. Introduction

### 1.1. Mental Health in Malaysia

Mental health is one of the leading causes of health-related burden among adolescents according to the Global Burden of Diseases, Injuries, and Risk Factors Study (GBD) [1]. The two mental health concerns reported in the study are depression and anxiety. The World Health Organisation (WHO) [2] has also reported that suicide is the fourth leading cause of death among those aged between 15 to 29 years. Additionally, the emergence of the COVID-19 pandemic has seen an increase in mental health concerns. Drastic changes, such as lockdowns, job losses, closure of schools and educational institutions, and economic breakdown, have led to an increase in the level of distress among children and young people in Malaysia [3]. Even prior to the pandemic, Malaysia had seen an increase in mental health concerns. According to the results attained from the National Health and Morbidity Survey [4], over one-third (29.2%) of Malaysians aged 16 and above experienced mental health problems. The mental health-related burden in Malaysia is approximately 37% of total disability [5]. According to the Minister of Health Malaysia, the Ministry’s psychosocial helpline has received 307,673 calls since March 2020, with 74.3% of the callers reaching out specifically to request emotional support as well as counselling due to anxiety, stress, and depression [6]. Furthermore, the National Health and Morbidity Survey 2019 [6] found that depression is prevalent among those aged 18 and above, with 2.3% being affected (approximately 500,000 people).

Mental health is included and discussed in the United Nations (UN) Sustainable Development Goals (SDG) under Goal 3: Good Health and Well-Being. One of the targets under Goal 3 is to “*reduce by one third premature mortality from non-communicable diseases through prevention and treatment and promote mental health and well-being*” by the year 2030 [7]. The UN took this step to acknowledge mental health issues and placed it as a priority when it comes to global development. Furthermore, Malaysia has also acknowledged the importance of addressing mental health issues by introducing the National Mental Health Strategic Plan 2020–2025. The plan states that there is a need to engage in initiatives to improve mental resilience among the young and elderly in order to prepare them for the difficulties and challenges that lie ahead [5]. Thus, the objective of this plan is as follows:

“*To promote mental health well-being, prevent mental disorders, provide care, enhance recovery, and reduce the mortality, morbidity, and disability for persons with mental health problems*” [5].

Knowing that mental health is a concerning issue, this study aimed to look into measures that could improve and curb mental health issues among youth in Malaysia using behavioural intervention technologies (BITs). Improvement of mental health refers to the enhancement of one’s mental well-being that will allow them to recognise their abilities, cope with life’s stresses, learn and perform successfully at work, and give back to their community [8]. BITs encompass the use of behavioural intervention strategies via the use of technological features in order to address three targets that support mental health, namely, cognitive, behavioural, and affective targets [9]. BITs provide the possibility of entirely new interventions, and these new interventions need to be evaluated when it comes to addressing mental health issues. This study used three underlying models, namely, the Ritterband model, Fogg behavioural model and persuasive system design, in order to attain a proposed set of behavioural intervention strategies.

A literature review was performed to further understand the topic in hand, and it led to the following research question:What is the role of behavioural intervention strategies (social distractions, peer motivation; ease of access to help; sense of belonging and mindfulness; mood changes; and thoughts, feelings, and action) in the improvement of mental health among youth in Malaysia?

Additionally, a research objective was established to allow the research direction to fall into place. The objective of this study was as follows:To examine the role of behavioural intervention strategies (social distractions; peer motivation; ease of access to help; sense of belonging and mindfulness; mood changes; and thoughts, feelings, and action) on the improvement of mental health among youth in Malaysia.

The study used three underlying models for intervention based on three design models, namely, the internet intervention model [10], Fogg behaviour model [11], and persuasive system design [12,13]. The underlying models intend to provide a context for the proposed research model and do not act as an exhaustive review and critique of theoretical models used to inform the development of BITs.

### 1.2. Three Intervention Models

#### 1.2.1. Internet Intervention Model

The internet intervention model was developed to produce change in behaviour and improvement in symptoms via effective internet interventions [10]. This intervention model features nine components that are set in stages. The model explains that *users*, who are influenced by *environmental* factors, tend to affect *website use*. Their website use, on the other hand, is influenced by the support that they receive as well as the *website characteristics*. An individual’s website use will then lead to a *change in behaviour* and *improvement of symptoms* through *mechanisms of change*. These improvements are maintained using *treatment maintenance*. This model developed by Ritterband et al. [10] is useful when it comes to developing websites or mobile applications as it specifies the important characteristics and elements required in order for internet intervention to be successful.

However, according to Mohr et al. [14], the internet intervention model does not exactly show how technological components could be mapped onto specifically detailed intervention goals. Mohr et al. [14] go on to say that because the model is not necessarily linear, Ritterband et al. [10] did not specify the nonlinear properties. These properties are important because the technologies used need to be able to receive, process, and react to the data attained from the user, the user’s environment, and the third parties involved (e.g., mental health therapists or mental health professionals) [14].

#### 1.2.2. Fogg Behaviour Model

The Fogg behaviour model [11] was developed to explain and understand how behaviour occurs. When there is an understanding of how behaviour occurs, it would assist in influencing a user’s behaviour. The model explains that one’s behaviour is the product of three constructs, namely, motivation, ability, and trigger. Fogg [11] emphasised that these three elements must occur at the same time as it will lead to behaviour. Motivation is an individual’s desire to act or behave in a certain manner, ability is when an individual has the capacity to perform the behaviour, and trigger is something that occurs when an individual is triggered to perform a behaviour using different or various cues [15]. Fogg [16] also explained that it is much easier to accept small changes compared to big changes when it comes to behaviour. These are referred to as “tiny habits”, whereby a series of planned small changes will result in the adoption of small habits that will lead to a desired behaviour. As the Fogg behaviour model is suitable for small behaviours, the model could be used as a tool in designing individual components when it comes to planning large intervention programs [14].

#### 1.2.3. Persuasive System Design

Persuasive system design [12,13] is a model that uses principles adopted and modified from the Fogg behaviour model [11]. Because users can be reached easily using the internet and other relevant technologies, this allows opportunities for persuasive interactions to occur. Persuasive systems may be utilised using computer-mediated persuasion or computer-human persuasion [12]. The persuasive system design model consists of four categories that consist of their own principles, namely, primary task support, dialogue support, system credibility, and social support. Primary task support consists of the reduction of complex behaviours, tunnelling whereby the system guides the users, tailoring, personalisation, self-monitoring, simulation, and rehearsal of behaviour [14]. Dialogue support consists of praise, rewards, reminders, suggestion, similarity, liking, and social role. As for system credibility support, it features principles such as trustworthiness, expertise, credibility, authority, and verifiability. Finally, social support incorporates social learning, social comparison, social facilitation, cooperation, competition, recognition, and normative influences [12]. Overall, this model could be used and adopted as part of the requirements for software functionality and intervention strategies.

### 1.3. Behavioural Intervention Strategies

One of the behavioural intervention strategies discussed in this study is social distractions. Social distraction occurs when people are not the targets of attention. Distractor faces show interference effects in visual search even under high perceptual load, suggesting mandatory processing [17]. According to researchers, it is important for young people to develop the ability to focus their attention voluntarily and control any possible distractions that they may face [18,19,20].

Another behavioural intervention strategy is the presence of peer motivation. Peers tend to be those who are within the same age, maturity, and level of perspective and who share social or demographic similarities [21,22]. On the other hand, motivation is described as an internal process and a condition within a person that desires changes, be it in terms of one’s self, environment, thoughts, feelings, or behaviour [23]. Thus, peer motivation is seen as someone with a shared set of similarities who pushes or encourages their peers to facilitate changes, either within themselves or the environment. Peer motivation or peer support in the context of mental health allows information to be shared and encourages behaviour modelling, which in turn could lead to greater autonomy when it comes to an individual’s recovery [24]. Peer motivation is important as the guidance given by these peers are able to influence the way individuals who are suffering from mental-health-related issues conduct themselves. According to Lubman et al. [25], peers are the most important source of support for youth and adolescents who are experiencing mental health issues.

Many of the challenges in providing mental health treatment have been eliminated with the help of technological solutions, and this has led to the ease of access to help. Individuals are able to quickly gain access to help using various digital health technologies, be it from the internet or mobile applications [26]. There are various interactive and psychoeducational applications for a wide range of mental health concerns available for download on the Google Play Store and Apple App Store [27]. People could easily download these applications and attain the necessary information that they need regarding their condition with just a press of a button. Additionally, the use of mobile devices allows individuals to overcome the physical barrier of distance when it comes to seeking help [28]. People are able to contact their local mental health careline or doctors and make appointments online without having to travel far away. On the other hand, the use of mobile symptom-tracking features allows easy access to help as the education mechanism integrated into the system could provide immediate feedback pertaining to the best options for the individual’s specific concerns [28].

The need to belong theory by Baumeister and Leary [29] explains that humans need to establish and maintain, at least at a minimum level, good interpersonal relationships. The authors go on to state that humans are naturally driven towards forming and sustaining belongingness [29]. According to Hagerty et al. [30], sense of belonging can be described as an individual’s personal participation in an environment or system that makes them feel that they are an integral part of the said environment or system. In Maslow’s hierarchy of needs, sense of belonging ranks third and has been identified as a basic human need [31,32]. Any disruption in a person’s relatedness or a lack of sense of belonging could lead to them having social, biological, and psychological disturbances; emotional distress; sense of loneliness; and mental illness [33]. However, studies have shown that those who experience sense of belonging are able to greatly reduce their levels of anxiety, depression, and loneliness [34,35].

Studies have also shown that mindfulness-based interventions help improve one’s well-being and reduce stress [36,37]. Mindfulness is described as the awareness of one’s thoughts, feelings, body sensations, and surroundings in the present moment [38]. There are two widely used mindfulness-based intervention programmes that are effective in improving common mental, social, and physical health conditions, namely, the mindfulness-based cognitive therapy (MBCT) [39] and mindfulness-based stress reduction (MBSR) [40].

At the adolescent stage, individuals are also prone to mood alterations alongside changes in attitude and behaviour [41]. An individual’s mood is controlled by a number of brain structures, chemical neurotransmitter systems, and networks [42]. According to Buchannan et al. [41], adolescents tend to experience more mood swings, restlessness, and intense moods compared to those in other stages of development. Additionally, the authors explained that adolescents will feel more self-conscious and more anxious. Mood changes may potentially reflect psychiatric or medical conditions, and if it prolongs for weeks, this indicates that it is time for an intervention.

The final intervention strategy is based on a psychological intervention called cognitive behavioural therapy (CBT), which emphasises a concept whereby an individual’s thoughts, feelings, and actions are interconnected [43]. CBT is a form of psychotherapy that uses a present-focused approach instead of focusing on an individual’s past problems when it comes to helping overcome depression, anxiety, eating disorders, and severe cases of mental illness [44]. CBT was initially developed to aid individuals facing depression and anxiety [45], but it is now applied to assist children and young adults who suffer from a wide range of mental-health-related problems. Nehra et al. [46] explains that the way one thinks greatly affects and influences the way one feels. Thus, when an individual is able to think differently, it will lead them to feeling and acting differently. That being said, CBT treatment typically involves an individual’ efforts to change their thinking patterns and behavioural patterns [44]. Kisely et al. [47] have suggested three mechanisms to classify this treatment. First, the intervention should involve the individual making relevant connections that link their thoughts, feelings, and actions while taking into consideration the target symptom. Second, the intervention should involve the correction of the individual’s reasoning bias, misperceptions, and irrational beliefs with regard to the target symptom. Finally, the intervention should involve the individual monitoring their own thoughts, feelings, and actions with regard to the target symptom and/or suggesting alternatives on how they could cope with the target symptom. According to Kazantzis et al. [48], the CBT treatment goals are to reduce distress, improve function, and enhance the quality and well-being of life.

Thus, based on the literature review, a research model was proposed as per Figure 1, and the following hypotheses were developed to understand the role of behavioural intervention strategies and its impact on improved mental health among youth in Malaysia:

**H_1_:** 
*There is a significant positive relationship between social distractions and improvement of mental health among youth in Malaysia.*


**H_2_:** 
*There is a significant positive relationship between peer motivation and improvement of mental health among youth in Malaysia.*


**H_3_:** 
*There is a significant positive relationship between ease of access to help and improvement of mental health among youth in Malaysia.*


**H_4_:** 
*There is a significant positive relationship between sense of belonging and mindfulness and improvement of mental health among youth in Malaysia.*


**H_5_:** 
*There is a significant positive relationship between mood changes and improvement of mental health among youth in Malaysia.*


**H_6_:** 
*There is a significant positive relationship between thoughts, actions, and feelings and improvement of mental health among youth in Malaysia.*


## 2. Methodology

This study was quantitative in nature, with data collected from respondents via questionnaire. The working population for this study was Malaysian youths between the ages of 18 to 23 years who were experiencing mental-health-related issues. The anonymity of the respondents was safeguarded throughout the data collection process. A five-point Likert scale affixed by measurements ranging from 1 (strongly disagree) to 5 (strongly agree) was used in the questionnaire. The questionnaire contained eight sections, namely, general questions (gender, age, and location); social distractions; peer motivation; ease of access to help; sense of belonging and mindfulness; mood changes; thoughts, feelings, and actions; and awareness of behavioural intervention strategies. A total of 103 individuals responded to the survey. The data collected was then analysed using the partial least squares (PLS) method using the SmartPLS 3 software. Table 1 depicts the research design components of this study.

## 3. Data Analysis

The data for this study were analysed using the SmartPLS 3 software. This software allows researchers to examine the inter-relationships between variables, whereby single or multiple regressions can be stated.

### 3.1. Measurement Model Evaluation

The measurement model evaluation was carried out to confirm the reliability and validity of the research model. The data obtained from the questionnaires were used to structure the measurement model of this study. Figure 2 illustrates the measurement model of this study.

The indicator reliability for the research model was assessed by referring to the factor loadings of each item. The factor loadings for each item should be valued above 0.708; however, there is a satisfactory threshold where the items do not necessarily have to be valued above 0.708. Table 2 depicts the factor loadings of each item valued between 0.621 to 0.899. Thus, this confirmed that indicator reliability was present. As for internal consistency reliability, this was determined by the composite reliability (CR), which should be valued above 0.70. Table 2 confirms that the CR for each construct was above 0.70, thus indicating that composite reliability was present. Additionally, the convergent validity was assessed using the average variance extracted (AVE). Constructs with AVE values above 0.50 signifies that there is a satisfactory degree of convergent validity. Table 2 affirms this as the AVE values for each construct were above 0.50.

Table 3 depicts the matrix derived to examine the discriminant validity of the research model. Based on the Fornell and Larcker criterion, items of a construct must show a stronger loading on its own construct compared to other constructs in the model. Each of the construct in the research model satisfied the criteria, thus confirming that there was sufficient discriminant validity.

### 3.2. Structural Model Evaluation

A structural model evaluation was conducted after assessing the reliability and validity of the measurement model. This evaluation was carried out to determine whether the hypotheses of the study were supported based on the data attained from the analysis. Figure 3 shows the structural model derived from the SmartPLS 3 software after nonparametric bootstrapping was conducted using a sample of 5000. Prior to assessing the path coefficient, an evaluation on the coefficient of determination (R^2^) was conducted to determine the structural model’s predictive power. The R^2^ value for this study was 0.492, which meant that 49.2% of the total variance in the dependent variable was explained by the independent variables. The R^2^ value here was considered to be moderate.

The path coefficient for this study is shown in Table 4. The beta value has to be at least 0.1 in order for it to make an impact to the research model, and the t-statistic has to be greater than 1.645 in order for it to be significant. Based on Table 4, mood changes (MC) and thoughts, feelings, and actions (TFA) had a significant positive influence on the improvement of mental health (IMH). However, ease of access to help (EAH), peer motivation (PM), sense of belonging and mindfulness (SBM), and social distractions (SD) did not have a significant positive influence on the improvement of mental health (IMH).

## 4. Discussion

The final results showed that only two out of six hypotheses were supported. Mood changes (MC) and thoughts, feelings, and actions (TFA) were deemed to be supported, while the rest, namely, ease of access to help (EAH), peer motivation (PM), sense of belonging and mindfulness (SBM), and social distractions (SD), were not supported. The reasons for these variables not being supported could be associated with the fact that awareness of behavioural intervention tools and strategies may still be low among Malaysian youth. This is largely due to the fact that mental health conditions, depression, and suicidal thoughts are still being viewed as social taboos in Asian society. Therefore, youth/children are not encouraged to talk about their mental health issues in public, and the taboo does not allow them to seek any assistance in handling these issues. Mood changes and thoughts, feelings, and action were significantly positively related to improved mental health as these are age-old techniques used by therapists, teachers, counsellors, and anyone interested in helping people suffering with mental health issues.

Additionally, theoretical frameworks guided by underpinning theories strengthen existing knowledge and assist in solving problems. They allow the objectives to be achieved by answering the research questions and connect the how and why of research. In Malaysia, the rise in youth suicide and incidence of youth who are easily affected by anxiety and depression clearly indicate the prevalence of mental health issues among this cohort. This study and framework sought to uncover and raise awareness of available strategies that can assist in curbing such issues at the infancy stage. The reason for lack of positive responses to certain factors is because there has been poor consideration of mental health in Malaysia, which is habitually viewed as taboo, making it difficult for youth to communicate and seek expert help to curb such thoughts and issues.

## 5. Conclusions

Mental health issues should be viewed seriously due to the growing number of cases of depression, anxiety, and suicidal thoughts among Malaysian youth. Recent epidemiological data in 2015 by the Malaysian Ministry of Health identified that the frequency of mental disorders among youth was 29%, representing a significant threefold increase compared to the 10% prevalence rate identified in 1996. It was also found that one in three Malaysians suffered from mental health issues, with the highest occurrence amongst youths. In Malaysia, culture plays a role because mental health is viewed as a taboo, and people tend to not speak about it as much as those in Western countries. Moreover, Western countries have easier access to mental healthcare facilities compared to Asian countries. Despite some methods or strategies being available, they have been little explored due to stigmatization and religious practices. Stigma and negative attitudes from the public pertaining to mental health is another barrier faced by Malaysians. It should not be a taboo to reach out for help, and the use of technology and various behavioural intervention strategies and tools should be encouraged to address mental health issues.

## Figures and Tables

**Figure 1 ijerph-19-15376-f001:**
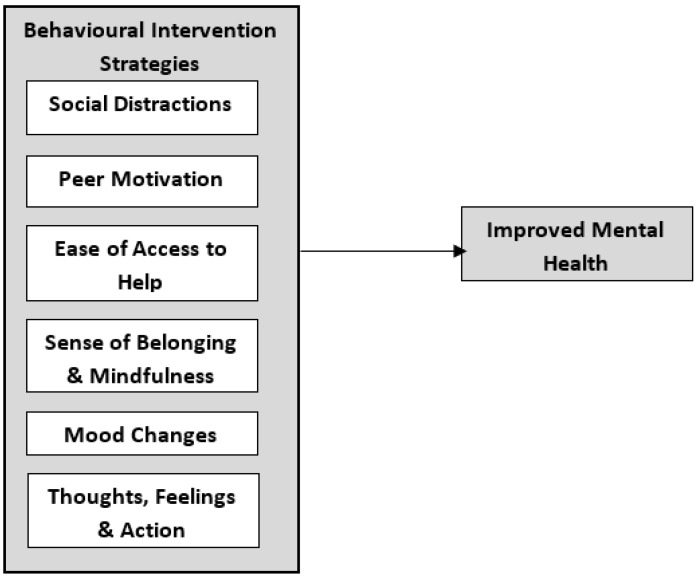
Theoretic model.

**Figure 2 ijerph-19-15376-f002:**
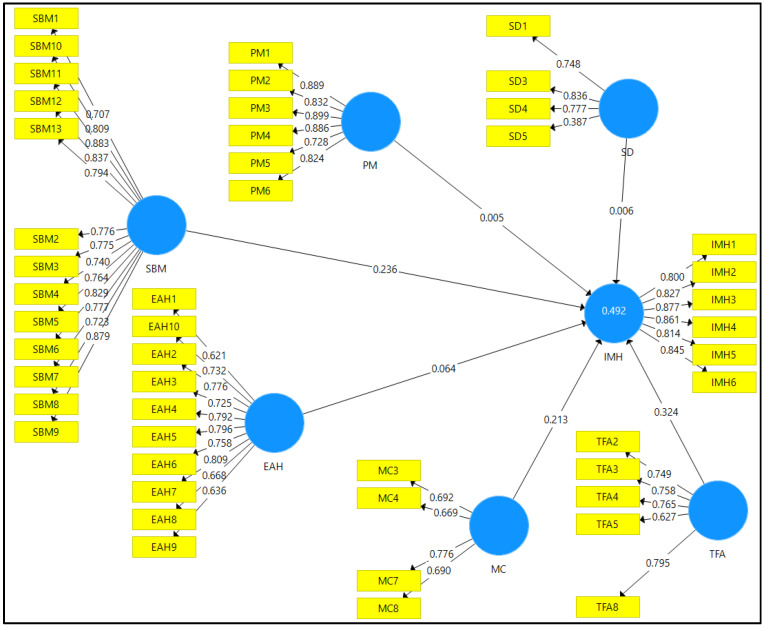
Measurement model.

**Figure 3 ijerph-19-15376-f003:**
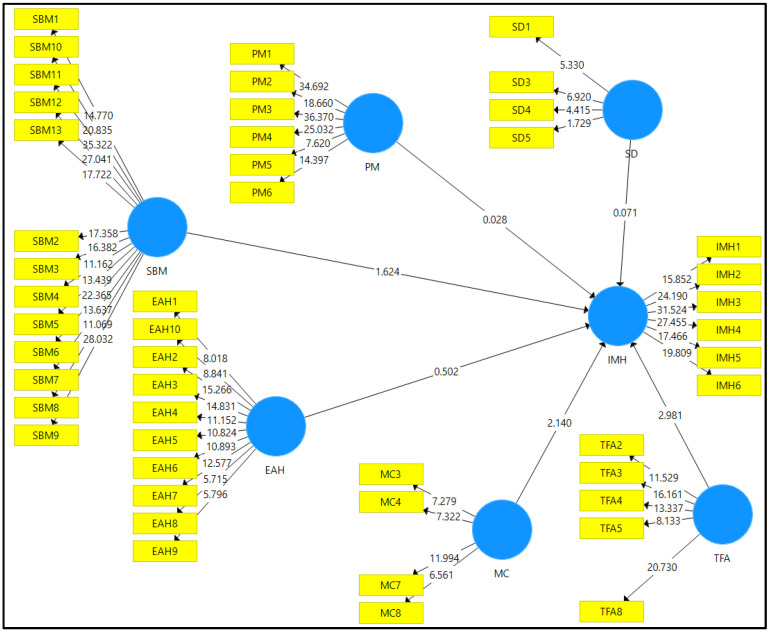
Structural model.

**Table 1 ijerph-19-15376-t001:** Research design components.

Research Design Component	Description	Rationalisation
Nature of study	Exploratory	The premise of this research was to determine the impact of behavioural intervention strategies in the improvement of mental health among youth in Malaysia.
Role of theory	To test the theory	A deductive approach was employed to test the hypothetical framework of this study. The research looked into the impact of behavioural intervention strategies, namely, social distractions; peer motivation; ease of access to help; sense of belonging and mindfulness; mood changes; and thoughts, feelings, and actions, on the improvement of mental health among youth in Malaysia.
Sampling process	Purposive sampling	The respondents were selected based on the following criteria: (i) aged between 18 to 23 years and (ii) Malaysian.
Data collection technique	Surveys	A questionnaire was prepared using Google Forms and distributed to Malaysians aged 18 to 23 years via social media platforms. A total of 103 responses were collected within a time period of two months.
Researcher interference	Minimal	During the distribution and collection of questionnaires, there was minimal interference to the work nature and activities of the respondents by the researchers.

**Table 2 ijerph-19-15376-t002:** Factor loadings, average variance extracted (AVE), and composite reliability for each construct and item.

Constructs	Items	Loadings	AVE	CR
Improved mental health	IMH1	0.800	0.702	0.934
	IMH2	0.827		
	IMH3	0.877		
	IMH4	0.861		
	IMH5	0.814		
	IMH6	0.845		
Ease of access to help	EAH1	0.621	0.539	0.921
	EAH10	0.732		
	EAH2	0.776		
	EAH3	0.725		
	EAH4	0.792		
	EAH5	0.796		
	EAH6	0.758		
	EAH7	0.809		
	EAH8	0.668		
	EAH9	0.636		
Mood changes	MC3	0.692	0.501	0.8
	MC4	0.669		
	MC7	0.776		
	MC8	0.690		
Peer motivation	PM1	0.889	0.714	0.937
	PM2	0.832		
	PM3	0.899		
	PM4	0.886		
	PM5	0.728		
	PM6	0.824		
Sense of belonging and mindfulness	SBM1	0.707	0.63	0.957
	SBM10	0.809		
	SBM11	0.883		
	SBM12	0.837		
	SBM13	0.794		
	SBM2	0.776		
	SBM3	0.775		
	SBM4	0.740		
	SBM5	0.764		
	SBM6	0.829		
	SBM7	0.777		
	SBM8	0.723		
	SBM9	0.879		
Social distractions	SD1	0.748	0.503	0.792
	SD3	0.836		
	SD4	0.777		
	SD5	0.387		
Thoughts, feelings, and action	TFA2	0.749	0.549	0.858
	TFA3	0.758		
	TFA4	0.765		
	TFA5	0.627		
	TFA8	0.795		

**Table 3 ijerph-19-15376-t003:** Discriminant validity matrix.

Constructs	EAH	IMH	MC	PM	SBM	SD	TFA
**EAH**	0.734						
**IMH**	0.458	0.838					
**MC**	0.475	0.547	0.708				
**PM**	0.572	0.512	0.574	0.845			
**SBM**	0.535	0.599	0.553	0.769	0.793		
**SD**	0.316	0.328	0.356	0.449	0.446	0.709	
**TFA**	0.502	0.621	0.52	0.507	0.635	0.367	0.741

**Table 4 ijerph-19-15376-t004:** Beta value, t-statistic, *p*-value, and hypothesis decision.

Constructs	Beta	T-Statistic	*p*-Value	Decision
EAH -> IMH	0.064	0.502	0.615	Not Supported
MC -> IMH	0.213	2.14	0.032	Supported
PM -> IMH	0.005	0.028	0.977	Not Supported
SBM -> IMH	0.236	1.624	0.105	Not Supported
SD -> IMH	0.006	0.071	0.943	Not Supported
TFA -> IMH	0.324	2.981	0.003	Supported

## Data Availability

The data attained from the questionnaire for this study are available at 10.6084/m9.figshare.21554670 (accessed on 15 November 2022). This file can be opened in Microsoft Excel.

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
