# Peer review of "Awareness of Behavioural Intervention Strategies in Curbing Mental Health Issues among Youth in Malaysia"

_ijerph, 2022, doi:10.3390/ijerph192215376_

Round 1

Reviewer 1 Report

I think Authors have done effort to analyze three underlying models for intervention in order to attain a set of behavioural intervention strategies. The data were analyzed with a valid methodology about possible inter-relationships between variables.

Despite this, after reading this manuscript, I have two main concerns:

1- I don't think it is necessary declare too many research questions, since can be confusive. It is more convenient focus on few specific topics.

2- the study described provides little additional content to the literature. Specifically, it is well established that mental health is a concerning issue. Probably it can find originallity about data on Malaysia' population

Related more minor comments:

- Can be usefull  develop a more concisely description about models and how models can ben usefull in the direction of menthal health. In this actual version I perceive a too broad description no well focused on the aims of the study

- The final results can be more developed with a more complete discussion

Author Response

Q1: I don't think it is necessary declare too many research questions, since can be confusive. It is more convenient focus on few specific topics.

The authors have decided to include only 1 research question after taking into consideration the reviewer's comment. We have changed it to: What is the role of behavioural intervention strategies (social distractions, peer motivation, ease of access to help, sense of belonging and mindfulness, mood changes, and thoughts, feelings and action) in the improvement of mental health among youth in Malaysia? 

Q2: The study described provides little additional content to the literature. Specifically, it is well established that mental health is a concerning issue. Probably it can find originality about data on Malaysia's population.

Reviewer 2 advised the authors to inlcude Literature Review under Introduction. Therefore, we have included the necessary data and description about mental health in Malaysia in Paragraph 1 of Introduction. 

Q3: Can be useful develop a more concisely description about models and how models can ben useful in the direction of mental health. In this actual version I perceive a too broad description no well-focused on the aims of the study.

We have added an additional paragraph in Discussion to address this.

Q4: The final results can be more developed with a more complete discussion.

We have added an additional paragraph in Discussion to address this.

Reviewer 2 Report

In the present manuscript, the authors examined the relationships between psychosocial factors and mental health improvement among young adults in Malaysia. The topic fits the scope of the journal and has potentials implications. However, there are several important issues in the paper.

1. The definition of mental health is vague, and there is no such term "mental health disorders". It should use either "mental health" or "mental disorder" in the paper.

2. Mental disorders include a wide range of psychiatric and neurological conditions. Those conditions are fundamentally different, and should be studied in separate analyses.

3. Please provide rationale why it is necessary to mental health issues in Malaysia. How is the status of mental health among youth different than other countries? What are implications of the findings from the current study? 

4. Literature Review is an inherent part of Introduction section, do not use a subsection.

5. What is "research model"? Does it mean "theoretic model"?

6. In hypotheses, please clearly state the direction of the predicted relationships.

7. What is the operational definition of "Improvement of Mental Health"?

8. Please provide descriptions of the questionnaires that were used in the study. Psychometric features of the questionnaires are necessary.

9. Based on the results of the analyses, there was no support for the statements in the Conclusion section.

Author Response

1. The definition of mental health is vague, and there is no such term "mental health disorders". It should use either "mental health" or "mental disorder" in the paper.

We have changed it to 'mental health' as this is our primary focus.

2. Mental disorders include a wide range of psychiatric and neurological conditions. Those conditions are fundamentally different, and should be studied in separate analyses.

The authors have taken note of this comment. Our focus for this study is on mental health and the behavioural intervention strategies instead of mental disorders.

3. Please provide rationale why it is necessary to mental health issues in Malaysia. How is the status of mental health among youth different than other countries? What are implications of the findings from the current study? 

Culture plays a role in mental health in Malaysia. Mental health is viewed as a taboo and people tend to not speak about it. We have included an explanation in Conclusion. 

4. Literature Review is an inherent part of Introduction section, do not use a subsection. 

We have included the literature review in the Introduction section.

5. What is "research model"? Does it mean "theoretic model"? 

We have changed it to 'theoretic model'.

6. In hypotheses, please clearly state the direction of the predicted relationships.

We have stated the direction of the predicted relationships by adding 'significant positive relationship' to the hypotheses.

7. What is the operational definition of "Improvement of Mental Health"? 

We have included the definition in Introduction: "The improvement of mental health refers to the enhancement of one’s mental well-being which will allow them to recognise their abilities, cope with life’s stresses, learn and perform successfully at work, and give back to their community."

8. Please provide descriptions of the questionnaires that were used in the study. Psychometric features of the questionnaires are necessary. 

There are no psychometric features as we only conducted a survey. The survey was conducted to gauge the behavioural aspect. In the Methodology section, we have provided the description of the questionnaire. 

9. Based on the results of the analyses, there was no support for the statements in the Conclusion section.

We have added explanations in the Discussion and Conclusion section.

Round 2

Reviewer 1 Report

The authors have satisfactorily addressed most of my concerns.The revised manuscript, in my opinon, is suitable for publication.